# UNA: Unifying Alignments of RLHF/PPO, DPO and KTO by a Generalized Implicit Reward Function

## Abstract

An LLM is pretrained on trillions of tokens, but the pretrained LLM may still generate undesired responses. To solve this problem, alignment techniques such as RLHF, DPO and KTO are proposed. However, these alignment techniques have limitations. For example, RLHF requires training the reward model and policy separately, which is complex, time-consuming, memory intensive and unstable during training processes. DPO proposes a mapping between an optimal policy and a reward, greatly simplifying the training process of RLHF. However, it can not take full advantages of a reward model and it is limited to pairwise preference data. In this paper, we propose **UN**ified **A**lignment (UNA) which unifies RLHF/PPO, DPO and KTO. Firstly, we mathematically prove that given the classical RLHF objective, the optimal policy is induced by a generalize implicit reward function. With this novel mapping between a reward model and an optimal policy, UNA can 1. unify RLHF/PPO, DPO and KTO into a supervised learning of minimizing the difference between an implicit reward and an explicit reward; 2. outperform RLHF/PPO while simplify, stabilize, speed up and reduce memory burden of RL fine-tuning process; 3. accommodate different feedback types including pairwise, binary and scalar feedback. Downstream experiments show UNA outperforms DPO, KTO and RLHF.

## 1 Introduction

LLMs are trained on extensive and diverse corpora, enabling them to develop robust language capabilities and a deep understanding of various contexts OpenAI et al. (2024); Anthropic (2024). However, during inference, LLM can generate undesired responses, which should be avoided. Supervised fine-tuning (SFT) though can improve an LLM on downstream tasks like question answering, it cannot solve these problems. To address these problems, alignment techniques like RLHF Ouyang et al. (2022) and DPO Rafailov et al. (2023) are proposed.

RLHF involves two stages of training from the SFT models as shown in part (b) of Figure 1. Firstly, it trains a reward model (RM) using a preference dataset consisting of tuples (input, desired response, undesired response). Next, during the RL fine-tuning stage, the policy generates responses to given prompts. These responses are evaluated by the reward model and then used to fine-tune the policy with RL through PPO. However, several problems exist in RLHF. First of all, there exists an overfitting problem in the training stage of the reward model. In addition, RL fine-tuning stage is inherently unstable due to the nature of RL. Lastly, RL increases memory requirements for elements like the policy, reference policy, reward model and value model.

DPO addresses these issues by creating a mapping between the reward model and the optimal policy, combining the RM and RL training stages into a single process as shown in part (c) of Figure 1. This approach simplifies the two-stage optimization into one stage, eliminating the need to train an explicit reward model, reducing memory costs, and transforming the unstable RL process into a stable binary classification problem. Given a prompt along with desired and undesired responses, the implicit rewards for both responses are calculated. The differences in these rewards are then used to optimize the policy. However, DPO has its own set of challenges. It cannot produce an explicit reward model and will require more preference data to fine-tune the LLM. Moreover, in RL, the pretrained RM can

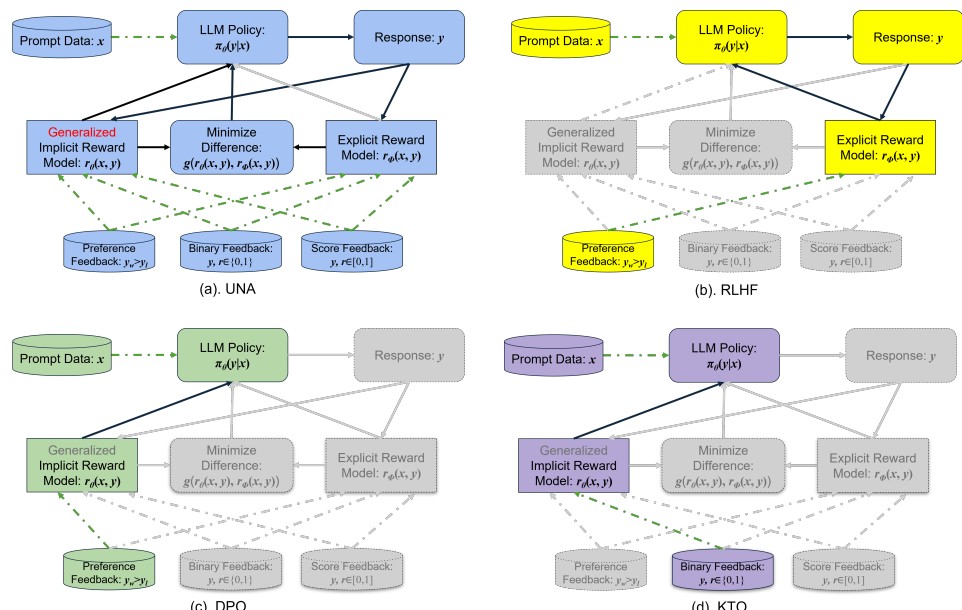

Figure 1: A figure comparison among (a). UNA, (b) RLHF, (c) DPO and (d) KTO. Each subfigure is composed of four types of data: "prompt data", "preference feedback", "binary feedback" and "score feedback", LLM policy, response, two reward models: "generalized implicit reward model" and "explicit reward model" and a module to minimize the difference between implicit and explicit rewards. The connection between data to other modules are utilizing green dash arrow, while others are connected by black solid arrow. All unused modules are grayed out. In part (b), RLHF firstly utilizes preference feedback to train the explicit reward model, and the use the evaluation provided by the explicit reward model to continuous optimize the policy in a online mode. In comparison, in part (c) and (d), DPO and KTO utilize preference feedback and binary feedback respectively to generate implicit reward to align LLM policy. However, in part (a), UNA can utilize different types of data to get generalized implicit and explicit rewards and minimize their differences to align LLM policy in online and offline modes.

provide accurate guidance for alignment, which is absent in DPO. In summary, DPO's efficiency in using preference data is lower compared to RLHF/PPO.

KTO extends DPO to handle binary data, such as thumbs up and thumbs down for desired and undesired responses as shown in the part (d) of Figure 1. However, there have not been work on alignment based on prompt, response and corresponding evaluation scores. In addition, there have not been a work that can unify RLHF/PPO, DPO and KTO to accommodate these different types of data. This work will address these problems.

In this work, we propose UNified Alignment (UNA) which unifies RLHF/PPO, DPO and KTO, and combines the benefits of them. Firstly, inspired by the derivation of DPO, we prove that based on the RLHF objective $\pi_\theta^*(y|x) = \max_{\pi_\theta} \mathbb{E}_{x \sim D} \left\{ \mathbb{E}_{y \sim \pi_\theta(y|x)}[r_\theta(x,y)] - \beta D_{\mathrm{KL}}\left(\pi_\theta(y|x)\|\pi_{\mathrm{ref}}(y|x)\right)\right\}$, the optimal policy can be induced by $r(x,y) = \beta \log\left(\frac{\pi_\theta(y|x)}{\pi_{\mathrm{ref}}(y|x)}\right) + f(x) + c$. It can be further simplified to $r(x,y) = \beta \log\left(\frac{\pi_\theta(y|x)}{\pi_{\mathrm{ref}}(y|x)}\right)$ when $f(x) = c = 0$. The condition $f(x) = c = 0$ indicates that the difference between implicit and explicit rewards is 0.

Based on the new generalized implicit reward function, UNA unifies RLHF/PPO, DPO and KTO into a supervised learning of minimizing the difference between an implicit reward and an explicit reward, where the explicit reward can come from human labelers, reward functions and LLMs as shown in part (a) of Figure 1. Given a prompt, the trained policy can firstly generate responses, and an implicit reward score can be calculated based on the previous Equation Then, the pair of prompt and response is evaluated by different evaluation tools to derive an explicit reward score. Provided the implicit and explicit reward score, a supervised learning problem like mean square error (MSE) can

be constructed to unify RLHF and DPO. Last but not least, for clarity, the unnormalized evaluation is termed as reward and the normalized evaluation is termed as score in this work.

With UNA, RLHF can be simplified through replacing the original RL fine-tuning stage, which is unstable, slow, and memory-intensive with a stable, efficient and memory friendly supervised learning. In addition, UNA can accommodate different types of data including pairwise feedback, binary feedback, score-based feedback. For pairwise data, we mathematically prove that UNA and DPO are equivalent. For binary data, thumb up (positive feedback) and thumb down (negative feedback) can be regarded as explicit rewards with reward scores of 1 and 0 respectively. With these derived implicit and explicit rewards, UNA can accommodate binary feedback. Lastly, for any types of unpaired data composed of a tuple, i.e., (prompt, response, score), UNA can be applied as well. Given the prompt and response, the implicit reward is firstly calculated, and then a supervised learning process is conducted to minimize the difference between the implicit reward and the explicit reward. In conclusion, UNA is a unified alignment framework for RLHF, DPO and KTO. It does not only simplify RLHF but also accommodates different types of data.

**The contributions of this paper are five-fold**:

1. Mathematically prove that based on the RLHF objective function, the optimal policy can be induced by the reward function $r(x, y) = \beta \log \left( \frac{\pi_\theta(y|x)}{\pi_{\text{ref}}(y|x)} \right) + f(x) + c$, which can simplified to $r(x, y) = \beta \log \left( \frac{\pi_\theta(y|x)}{\pi_{\text{ref}}(y|x)} \right)$ when $f(x) = c = 0$.

2. Propose UNA which unifies RLHF/PPO with DPO into a supervised learning of minimizing the difference between implicit reward and explicit reward.

3. Propose UNA that outperforms RLHF/PPO while simplifies, stabilizes, speeds up and reduces memory burden of RL fine-tuning process.

4. Propose UNA that can accommodate different types of data: pairwise feedback, binary feedback, score-based feedback on both online and offline mode from different evaluation methodologies including human labeling, reward models and LLMs.

5. Evaluate the performance of UNA on downstream tasks and compare it with DPO, KTO and RLHF/PPO to show its benefits.

## 2 METHODOLOGY: UNA

In this section, we will starts with some review of RLHF/PPO and DPO. Then, we will introduce UNA and derive a general loss function and its four applications: 1. Equivalence to DPO for pairwise preference dataset; 2. Improvement KTO for binary feedback; 3. RM / LLM distillation using reward from RM / LLM; 4. Simplification of RLHF in RL fine-tuning stage.

### 2.1 RLHF/PPO

After the SFT phase, the RLHF using PPO consists of two main stages: reward model training and reinforcement learning fine-tuning.

During the reward model training process, an explicit reward model is trained to predict a reward score $r_\phi(x, y)$ based on a given prompt $x$ and response $y$. This training utilizes pairwise preference data in the form of tuples, specifically $(x, y_w, y_l)$, where $y_w$ represents the desired response and $y_l$ represents the undesired response. Initially, the probability of $y_w$ being preferred over $y_l$, denoted as $P_\phi(y_w > y_l|x)$, is calculated based on their respective reward scores $r_\phi(x, y_w)$ and $r_\phi(x, y_l)$ through the Bradley-Terry (BT) model as shown in Equation 1, which provides a probabilistic framework for comparing the preferences between the two responses.

$$P_\phi(y_w > y_l|x) = \frac{e^{r_\phi(x,y_w)}}{e^{r_\phi(x,y_w)} + e^{r_\phi(x,y_l)}} = \sigma(r_\phi(x, y_w) - r_\phi(x, y_l)) \tag{1}$$

Given a pre-collected pairwise dataset where humans have selected the desired and undesired responses from two candidates, we have $P(y_w > y_l|x) = 1$ and $P(y_w < y_l|x) = 0$. To train an

effective reward model, we minimize the cross-entropy loss between the predicted probabilities and the human-labeled probabilities as shown in Equation 2. Once the cross-entropy loss is minimized, the training of the reward model is complete.

$$L_{\text{RM}}(\pi_\theta) = -\mathbb{E}_{(x,y_w,y_l)\sim D}\left[\log(\sigma(r_\phi(x,y_w) - r_\phi(x,y_l)))\right] \tag{2}$$

The second stage of RL fine-tuning has two primary goals. The first goal is to maximize the pretrained explicit reward function $r_\phi(x,y)$ to ensure the policy aligns with reward model. To prevent reward hacking, the KL divergence from the initial policy $\pi_{\text{ref}}(y|x)$ is incorporated. The overall objective of RL fine-tuning is detailed in Equation 3.

$$\pi_\theta^*(y|x) = \max_{\pi_\theta} \mathbb{E}_{x\sim D}\left\{\mathbb{E}_{y\sim\pi_\theta(y|x)}\left[r_\phi(x,y)\right] - \beta D_{\text{KL}}\left(\pi_\theta(y|x)\|\pi_{\text{ref}}(y|x)\right)\right\} \tag{3}$$

Several limitations exist in RLHF. To begin with, the reward model may suffer from overfitting during training, which can adversely affect the RL fine-tuning process. Then, unlike traditional supervised learning, RL does not have explicit labels for each prompt and response. To address this, the authors employed PPO to optimize the RL objective. However, even with PPO, RL training can still be unstable. Additionally, RLHF with PPO necessitates the use of a policy, reference policy, reward model, and value model, which significantly increases memory requirements, especially for LLMs. These limitations constrain the practical application of RLHF.

## 2.2 DPO

In RLHF, the trained reward model can suffer from overfitting, and RL fine-tuning is notorious for its instability and memory intensity. To address these challenges, the authors of DPO discover that the optimal policy is induced by Equation 4, based on the objective function in Equation 3. Here, $Z(x) = \sum_y \pi_{\text{ref}}(y|x)e^{\left(\frac{1}{\beta}r_\theta(x,y)\right)}$, where $r_\theta(x,y)$ represents the implicit reward function.

$$r_\theta(x,y) = \beta\log\left(\frac{\pi_\theta(y|x)}{\pi_{\text{ref}}(y|x)}\right) + \beta\log Z(x) \tag{4}$$

With the derived implicit reward model, it can be plugged into the reward model training process of RLHF in Equation 2 where $Z(x)$ gets cancelled. Eventually, the loss function for DPO is derived as shown in Equation 5.

$$L_{\text{DPO}}(\pi_\theta) = -\mathbb{E}_{(x,y_w,y_l)\sim D}\left\{\log\left[\sigma\left(\beta\log\left(\frac{\pi_\theta(y_w|x)}{\pi_{\text{ref}}(y_w|x)}\right) - \beta\log\left(\frac{\pi_\theta(y_l|x)}{\pi_{\text{ref}}(y_l|x)}\right)\right)\right]\right\} \tag{5}$$

By optimizing the loss function in DPO, we can eliminate the need for an explicit reward model and combine the two stages of RLHF into a single, streamlined process, greatly simplifying the RLHF/PPO workflow. However, DPO has several limitations. First, $Z(x)$ cannot be directly estimated, which means only pairwise preference data can be utilized, making single-prompt data unusable during the RL fine-tuning stage. Additionally, while pairwise preference data are typically used only in the reward model stage, DPO requires them throughout, leading to inefficient use of precollected pairwise data. In comparison, after reward model training, it can be applied to prompt data, which are much easier to obtain compared with pairwise data. Lastly, in the RL stage in RLHF, reward model can provide more detailed evaluations of the generated responses. However, DPO cannot offer this level of granularity during training.

## 2.3 UNA

Inspired by the idea of DPO, we aim to establish a new relationship between the reward model and the optimal policy for a unified alignment framework including RLHF/PPO, DPO and KTO on different types of data. By adhering to the same objective outlined in RLHF (Equation 3), we can formulate a novel connection between the implicit reward function and the optimal policy, as shown in Equation 6. The derivation can be found in Section **??**. In the special case where $f(x) = 0$ and $c = 0$, it is further simplified.

$$r_\theta(x, y) = \beta \log \left( \frac{\pi_\theta(y|x)}{\pi_{\text{ref}}(y|x)} \right) + f(x) + c$$

$$= \beta \log \left( \frac{\pi_\theta(y|x)}{\pi_{\text{ref}}(y|x)} \right) \text{ when } f(x) = 0 \text{ and } c = 0$$

(6)

The optimal implicit reward formulation in Equation 6 implies that we can transform the original unstable, memory-expensive RL training process into a reward function optimization problem, i.e., a stable and memory-efficient supervised learning process. Explicit rewards can be derived from multiple methods including 1. human labeling, 2. pretrained LLMs and 3. reward models. Eventually, the RL fine-tuning process is transformed into a general minimization problem between explicit reward $r_\phi(x, y)$ and implicit reward $r_\theta(x, y)$ as shown in Equation 7 where $g(x, y)$ refers to a general function that measure the difference between $x$ and $y$ like MSE.

$$L_{\text{UNA-reward}}(\pi_\theta) = \mathbb{E}_{x \sim D} \mathbb{E}_{y \sim \pi_\theta(\cdot|x)} [g(r_\phi(x, y), r_\theta(x, y))]$$

(7)

When applying an LLM for evaluation, the scores lie within a specific range, such as [0, 100]. These scores can be easily normalized to the interval [0, 1]. However, the implicit reward function can span from negative to positive infinity. To normalize implicit reward, the implicit score function, denoted as $s_\theta(x, y)$, can be derived as shown in Equation 8. For clarity, the unnormalized evaluation is termed as reward and the normalized evaluation is termed as score.

$$s_\theta(x, y) = \sigma[r_\theta(x, y)] = \sigma \left[ \beta \log \left( \frac{\pi_\theta(y|x)}{\pi_{\text{ref}}(y|x)} \right) \right]$$

(8)

Given the implicit and explicit score functions, an equivalent general loss for UNA can be shown in Equation 9. The normalized general loss function is more stable and will be used for experiments in this study.

$$L_{\text{UNA-score}}(\pi_\theta) = \mathbb{E}_{x \sim D} \mathbb{E}_{y \sim \pi_\theta(\cdot|x)} [g(s_\phi(x, y), s_\theta(x, y))]$$

(9)

Based on this general loss function using the new implicit reward function, UNA can be utilized in multiple conditions: 1. Equivalence to DPO for pairwise preference dataset 2. Improvement over KTO for binary feedback 3. RM / LLM distillation using reward from teacher RM / LLM outperforming DPO and KTO 4. Improvement over RLHF in RL fine-tuning stage: simplify PPO with a supervised learning process. Figure 2 shows how UNA is applied to different types of data and simplifies RLHF.

### 2.3.1 UNA: EQUIVALENT TO DPO FOR PAIRWISE DATASET

For pairwise dataset, the implicit rewards of desired and undesired responses can be derived as shown in part (a) of Figure 1. Then, LLM policy is aligned by maximizing the difference of implicit rewards between desired and undesired responses. The loss function of UNA for pairwise dataset is shown in Equation 10.

$$L_{\text{UNA-pair}}(\pi_\theta) = -\mathbb{E}_{(x, y_w, y_l) \sim D} \left( r_\theta(x, y_w) - r_\theta(x, y_l) \right)$$

$$= -\mathbb{E}_{(x, y_w, y_l) \sim D} \left[ \beta \log \left( \frac{\pi_\theta(y_w|x)}{\pi_{\text{ref}}(y_w|x)} \right) - \beta \log \left( \frac{\pi_\theta(y_l|x)}{\pi_{\text{ref}}(y_l|x)} \right) \right]$$

(10)

It is equivalent to DPO as the loss function is the same as long as $f(x) = \log[\sigma(x)]$ is applied to the difference of implicit rewards of desired and undesired responses: $L'_{\text{UNA-pair}}(\pi_\theta) = L_{\text{DPO}}(\pi_\theta) = -\mathbb{E}_{(x, y_w, y_l) \sim D} \{\log[\sigma(r_\phi(x, y_w) - r_\phi(x, y_l))]\}$

### 2.3.2 UNA: IMPROVEMENT OVER KTO FOR BINARY FEEDBACK

For binary preference, the positive and negative feedback can be transformed to explicit scores. Positive or 'thumb up' data can be assigned an explicit reward score of 1, i.e., $s_\phi(x, y_w) = 1$.

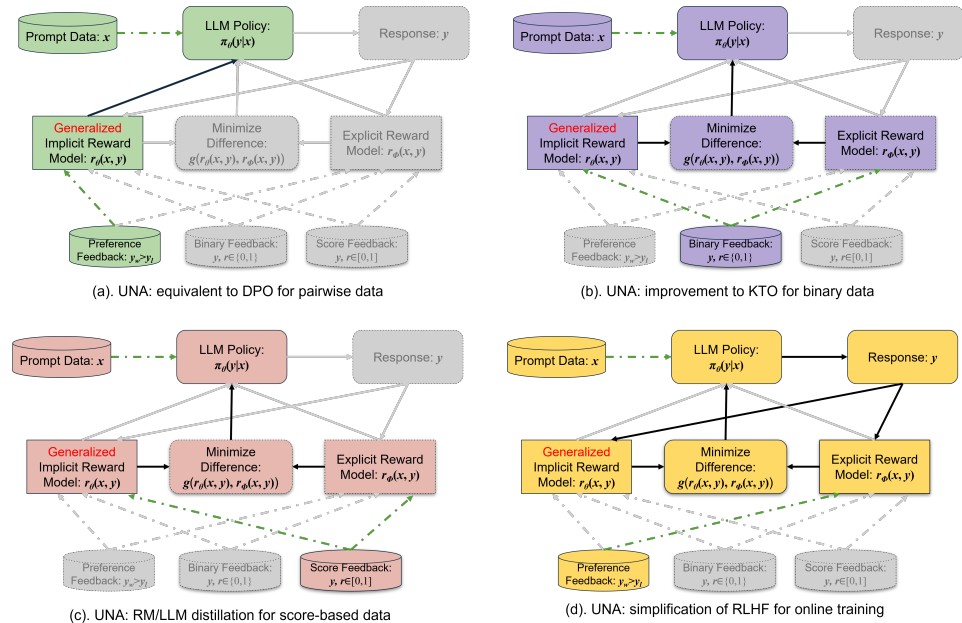

Figure 2: The four applications of UNA: (a). equivalent to DPO for pairwise data, (b). improvement over KTO for binary data, (c). RM/LLM distillation for score-based data, (d). simplification of RLHF for online training. The same modules are utilized as in Figure 1, and unused modules are grayed out. For part (a), the same steps as DPO will be utilized. For part (b), (c), (d), from the different types of data including pairwise, binary and score-based feedback, implicit and explicit rewards are firstly gathered. Then, the difference between implicit and explicit rewards is minimized like MSE loss function to align the LLM policy. More details can be found in Section 2.3.

In contrast, negative or 'thumb down' data can be assigned an explicit reward score of 0, i.e., $s_\phi(x, y_l) = 0$. After that, similar procedures to DPO will be conducted to estimate implicit reward and minimize the difference between implicit and explicit rewards as shown in part (b) of Figure 2.

Because the explicit feedback is binary, i.e., score rather than reward, implicit score should be utilized. Based on the implicit and explicit scores, multiple loss functions can be designed using mean square error (MSE) in Equation 11 and binary cross entropy (BCE) in Equation 12. As a result, UNA can be utilized to improve KTO for binary feedback data.

$$
\begin{aligned}
L_{\text{UNA-binary-MSE}}(\pi_\theta) &= -\mathbb{E}_{(x,y)\sim D}[(s_\theta(x,y) - s_\phi(x,y))^2] \\
&= -[\mathbb{E}_{(x,y_w)\sim D}(s_\theta(x,y) - 1)^2 + \mathbb{E}_{(x,y_l)\sim D}(s_\theta(x,y) - 0)^2]
\end{aligned}
\tag{11}
$$

$$
\begin{aligned}
L_{\text{UNA-binary-BCE}}(\pi_\theta) &= -\mathbb{E}_{(x,y)\sim D}[L_{\text{BCE}}(s_\theta(x,y), s_\phi(x,y))] \\
&= -[\mathbb{E}_{(x,y_w)\sim D}\log(s_\theta(x,y)) + \mathbb{E}_{(x,y_l)\sim D}\log(1 - s_\theta(x,y))]
\end{aligned}
\tag{12}
$$

### 2.3.3 UNA: LLM / RM DISTILLATION

Researchers have utilized LLM and RM to evaluate responses by outputting scores and rewards according to predefined standards. If the score and reward evaluations are accurate enough, they can be extra information to utilize for alignment. When the tuple type of data (prompt, response, score) is provided, the prompt and response are utilized to calculate implicit reward as shown in Equation 6, and the score is utilized as the explicit reward as shown in part (c) of Figure 2. The last step is the minimization of implicit and explicit rewards. However, the explicit reward and score from reward model and LLM are not binary, and as a result, MSE can be used as the loss function, excluding BCE. After normalization, the loss function for UNA using LLM / RM distillation is shown in Equation 13. When LLMs are utilized for evaluation, it can be regarded as an offline version of RLAIF.

$$L_{\text{UNA-LLM-distill}}(\pi_\theta) = -\mathbb{E}_{(x,y)\sim D}[(s_\theta(x,y) - s_\phi(x,y))^2] \tag{13}$$

### 2.3.4 UNA: SIMPLIFICATION OF RLHF

When utilizing reward model for online evaluation, UNA will greatly simplify RL fine-tuning stage of RLHF/PPO with superior performances as shown in part (d) of Figure 2. Assuming the reward model has already been trained, the focus now shifts exclusively to the RL fine-tuning stage. Prompts are firstly sent to an LLM for online response generation and implicit reward estimation. Then, the prompt and response are sent to the reward model for explicit reward estimation. The last step minimize the differences between implicit and explicit rewards to align the LLM policy. Eventually, the original RL objective in Equation 3 can be transformed to difference minimization like MSE of implicit reward and explicit reward or scores as shown in Equation 13

UNA has several benefits over PPO in RL fine-tuning stage. First of all, it transforms the original unstable RL problem into a stable supervised learning problem by minimizing the difference between implicit and explicit rewards. In addition, UNA removes the necessary of value model in PPO, and partially reduce the burden of memory cost. Then, the computation cost of MSE is much lower compared with the multiple terms in PPO to maintain performance, and as a result, UNA will speed up the training process. Lastly, UNA outperforms RLHF/PPO on downstream tasks.

## 3 EXPERIMENTS

In this section, we will evaluate UNA on three types of experiments: improvements over DPO in pairwise feedback and KTO in binary feedback, RM/LLM distillation for score-based response and simplification to online RLHF. For the first two tests, `mistralai/Mistral-7B-v0.1` Jiang et al. (2023) is utilized as the policy model, and the `HelpSteer2` dataset Nvidia et al. (2024) is utilized as the alignment data, which have a prompt, chosen and rejected responses with corresponding scores. The evaluation scores in `HelpSteer2` are labeled by human from the perspectives of 1. *helpfulness*, 2. *correctness* 3. *coherence* 4. *complexity* and 5. *verbosity*, and the combined score is computed as: $0.65 \times$ helpfulness $+ 0.8 \times$ correctness $+ 0.45 \times$ coherence, following **?**. Low rank adaptation (LoRA) Hu et al. (2021) is employed during the fine-tuning process with $r = 32$, where $r$ denotes the ranks used in LoRA. For $\beta$, UNA-binary utilizes 0.01 and DPO, KTO and UNA-score utlizes 0.03. For learning rate, UNA-score employs 3e-5 while others utilize 5e-6.

For the simplification of RLHF experiments, due to the computation availability and LoRA is not supported in PPOv2, `Qwen/Qwen2-1_5B` Yang et al. (2024a) is utilized as the policy model and `Ray2333/GRM-Gemma-2B-rewardmodel-ft` Yang et al. (2024b) is utilized as the reward model. The prompts of the same `Helpsteer2` dataset are utilized excluding the prompts longer than 1000 tokens. These experiments shows that UNA outperforms RLHF. For $\beta$, RLHF utilizes 0.05, while UNA uses 0.03, with both approaches employing the same learning rate of 5e-6.

After alignment, the old and new HuggingFace Open LLM Leaderboards Beeching et al. (2023); Fourrier et al. (2024) are both utilized to measure the performance. The new Open LLM Leaderboard contains 6 tasks: bbh Suzgun et al. (2022), gpqa Rein et al. (2023), mmlu-pro Wang et al. (2024), musr Sprague et al. (2024), ifeval Zhou et al. (2023) and math-hard Hendrycks et al. (2021b). For all tasks, the average scores of all tasks are reported. On the other hand, the old Open LLM Leaderboard contains other 6 tasks: gsm8k Cobbe et al. (2021), truthful-qa Lin et al. (2022), winograde Sakaguchi et al. (2019), arc-challenge Clark et al. (2018), hellaswag Zellers et al. (2019) and mmlu Hendrycks et al. (2021a). In this work, the average match rate in gsm8k, mc2 in truthful-qa, acc in winograde, acc-norm in arc-challenge, acc-norm in hellaswag and acc in mmlu will be reported. In addition to evaluating the model's selection capabilities, MT-Bench and Alpaca-eval will also be used to assess the model's ability to generate text responses, rather than selecting from predefined candidate answers.

### 3.1 UNA: IMPROVEMENTS OVER DPO & KTO

For binary feedback, borrowing the idea of KTO, the chosen responses are regarded as desired response with score "+1" and rejected response are regarded as undesired response with score "0". In

this way, the explicit scores are obtained. The generalized implicit rewards are firstly transformed into implicit reward scores, i.e., $s_\theta(x, y) = \sigma[r_\theta(x, y)] = \sigma\left[\beta \log\left(\frac{\pi_\theta(y|x)}{\pi_{\text{ref}}(y|x)}\right)\right]$. Then, different loss functions including BCE and MSE are utilized to minimize the differences between implicit and explicit reward scores.

In score-based feedback for `HelpSteer2`, human annotators assign initial scores to each metric, ranging from 0 to 4. These scores are then normalized to a 0 to 1 scale. The normalized scores are subsequently weighted, and the resulting weighted scores are used as explicit feedback to align the LLM. The same implicit reward scores as before are utilized. Because the explicit reward score is a continuous variable in the interval [0, 1], MSE is utilized as the loss function.

The results are shown in Table 1 for the new Open LLM Leaderboard and Table 2 for the old Open LLM Leaderboard. The highest scores for each metric and average are stressed in bold. For binary data, UNA performs better than all the baselines do on both Leaderboards. Lastly, for score-based feedback, it further improves over UNA-binary on both Leaderboards, as more information is provided. Consequently, when precise score-based information is available, it is recommended to leverage it.

| Method | bbh | gpqa | mmlu-pro | musr | ifeval | math-hard | Average |
|---|---|---|---|---|---|---|---|
| Mistral | 44.11 | 29.53 | 30.11 | 41.79 | 23.22 | 2.92 | 28.61 |
| DPO (UNA-pairwise) | **44.5** | 28.48 | 30.41 | 39.25 | 26.3 | 2.25 | 28.53 |
| KTO | 44.46 | 29.51 | 30.43 | 40.45 | 24.18 | 2.34 | 28.56 |
| UNA-binary (MSE) | 44.32 | 29.86 | 30.54 | 39.11 | 26.1 | **3.32** | 28.88 |
| UNA-binary (BCE) | 44.43 | 29.42 | **30.73** | 39.51 | 26.49 | 2.99 | 28.93 |
| UNA-score (MSE) | 43.53 | **30.25** | 29.72 | **42.01** | **37.25** | 2.77 | **30.92** |

Table 1: The comparison of UNA with DPO, KTO considering pairwise, binary and score-based data on new Open LLM Leaderboard

| Method | gsm8k | truthful-qa | winograde | arc | hellaswag | mmlu | Average |
|---|---|---|---|---|---|---|---|
| Mistral | 38.02 | 42.58 | 77.58 | 61.43 | 83.44 | 62.51 | 60.93 |
| DPO (UNA-pairwise) | 40.22 | 44.75 | 79.16 | 62.88 | 84.42 | 62.15 | 62.26 |
| KTO | **41.63** | 47.72 | 78.14 | 62.29 | 84.21 | 62.46 | 62.74 |
| UNA-binary (MSE) | 40.87 | 48.23 | 79.48 | **63.23** | 84.57 | 62.34 | 63.12 |
| UNA-binary (BCE) | 40.41 | 48.33 | 79.4 | 63.14 | **84.6** | 62.48 | 63.06 |
| UNA-score (MSE) | 40.41 | **55.09** | **80.27** | **63.23** | 84.52 | **62.56** | **64.35** |

Table 2: The comparison of UNA with DPO, KTO considering pairwise, binary and score-based data on old Open LLM Leaderboard

We also conducted evaluations on both MT-Bench Zheng et al. (2023) and AlpacaEval Li et al. (2023). UNA-binary (MSE) achieves the highest performance on MT-Bench, while UNA-score (MSE) leads on AlpacaEval, as seen in Table 3. The performance results from LLM Leaderboards, MT-Bench, and AlpacaEval clearly demonstrate the advantages of UNA over DPO and KTO.

| Method | MT-Bench | Alpacaeval LC WR |
|---|---|---|
| Mistral | 3.15 | 0.31 |
| DPO (UNA-pairwise) | 6.1 | 3.67 |
| KTO | 5.99 | 4.46 |
| UNA-binary (MSE) | **6.78** | 5.54 |
| UNA-binary (BCE) | 6.23 | 7.41 |
| UNA-score (MSE) | 6.72 | **8.78** |

Table 3: The comparison of UNA with DPO, KTO considering pairwise, binary and score-based data on MT-Bench and AlpacaEval using `HelpSteer2` as fine-tuning data

## 3.2 UNA: Improvement and Simplification on online RLHF

For the comparison between RLHF and UNA, only prompts of `HelpSteer2` are utilized. In RLHF, the prompts are sent to the policy for response generation, to the reward model for reward estimation

and to the policy for update through PPO. In comparison, in UNA, the prompts are sent to the policy for response generation and implicit reward estimation, to the reward model for explicit reward estimation and to the policy for update through difference minimization like MSE between implicit and explicit rewards.

The comparison between RLHF and UNA is shown in Table 4 and Table 5. UNA outperforms RLHF in 12 out of 14 tasks. Overall, UNA beats RLHF in both Open LLM Leaderboards. More comparison of RLHF with UNA on MT-Bench and AlpacaEval can be found in Table 6. The performance results from LLM Leaderboards, MT-Bench, and AlpacaEval clearly demonstrate the superiority of UNA over RLHF.

| Method | bbh | gpqa | mmlu-pro | musr | ifeval | math-hard | Average |
|--------|------|------|----------|------|--------|-----------|---------|
| Qwen2-1.5B | 35.46 | 25.16 | **25.56** | 36.85 | 22.2 | 5.4 | 25.11 |
| RLHF | 35.57 | **26.7** | 25.17 | 36.84 | 22.37 | **5.48** | 25.36 |
| UNA | **36.03** | 25.62 | 25.3 | **38.32** | **24.78** | 5.4 | **25.91** |

Table 4: The comparison of UNA with RLHF using `HelpSteer2` prompts on new Open LLM Leaderboard

| Method | gsm8k | truthful-qa | winograde | arc | hellaswag | mmlu | Average |
|--------|-------|-------------|-----------|------|-----------|------|---------|
| Qwen2-1.5B | **57.92** | 45.93 | 66.06 | 43.94 | 66.72 | **55.82** | 56.07 |
| RLHF | 57.2 | 46.93 | 64.88 | 42.83 | 66.56 | 55.67 | 55.68 |
| UNA | 57.36 | **47.08** | **65.27** | **44.28** | **66.98** | 55.78 | **56.13** |

Table 5: The comparison of UNA with RLHF using `HelpSteer2` prompts on old Open LLM Leaderboard

| Method | MT-Bench | AlpacaEval LC WR |
|--------|----------|------------------|
| Qwen | 4.63 | 1.06 |
| RLHF | 2.87 | 0.66 |
| UNA | **5.02** | **1.63** |

Table 6: The comparison of UNA with RLHF using `HelpSteer2` prompts on MT-Bench and AlpacaEval

Last but not the least, because UNA has transformed RLHF from an RL task into a supervised learning problem and got rid of the value model, the memory usage and time cost are greatly reduced for training. The training time for 20,000 steps with 8 80G A100 GPUs is around 8 hours for RLHF and 3.5 hours for UNA with the same batch size. The speed improvement of UNA over RLHF is significant, and these advantages can be amplified with a larger batch for UNA, which is impractical for RLHF due to its higher memory costs. In conclusion, with improved performances, a more stable loss function, memory-efficient and faster training, UNA outperforms RLHF from multiple perspectives.

## 4 RELATED WORK

The field of LLM has been greatly revolutionized with billions of parameters, trillions of tokens in parallel during the pretraining stage OpenAI et al. (2024); Anthropic (2024); Team et al. (2023). After pretraining, SFT will be applied to enhance its capability on downstream tasks. However, both pretraining and SFT can not solve the bias and ethic problem of LLM as they exist in the pretriaing data OpenAI et al. (2024). To solve this problem, RLHF with PPO Ouyang et al. (2022); Bai et al. (2022a) have been proposed, and it is the mostly accepted method to align LLM including GPT and Claude. However, lots of problems exist for RLHF/PPO including large memory burden, unstability of RL and multiple stages of training, i.e. RM training and RL fine-tuning Rafailov et al. (2023). To decrease the cost of human labelling, AI feedback can be utilized to replace human feedback, which will be termed as RLAIF Bai et al. (2022b); Lee et al. (2023). RLOO considers PPO an overkill for LLM alignment as LLM has been pretrained, and RLOO should be good enough Ahmadian et al. (2024).

To simplify RLHF, DPO is proposed to map the optimal policy and reward model, and the two stages can be merged into one step Rafailov et al. (2023). This process transforms the initial unstable RL into a binary cross entropy problem. DPOP Pal et al. (2024) mathematically prove that during DPO, the reward of desired responses will go down and proposed a maximum term to prevent the rewards of desired responses going down. IPO discovered that under nearly deterministic condition between desired and undesired responses, the effectiveness of the KL divergence constraint imposed by $\beta$ diminished, potentially leading to overfitting, and they proposed a new loss term to prevent this problem Azar et al. (2023). sDPO proposed to divide given dataset into splits and use these splits to sequentially align the model will achieve performance than using all of them at once Kim et al. (2024). Iterative DPO argued that LLM can be both response generator and evaluator so that it can iterate and improve it continuously Yuan et al. (2024); Xu et al. (2024). TDPO provided an idea to provided reward to each token generation Rafailov et al. (2024); Zeng et al. (2024).

There have also been some works on merging SFT with alignment. ORPO proposed a new loss function to increase the ratio of desired responses over undesired responses to realize the goal of both SFT and alignment Hong et al. (2024). PAFT proposed to conduct SFT and alignment in parallel and merge them together afterward Pentyala et al. (2024). Some works, i.e., R-DPO Park et al. (2024) and SimPO Meng et al. (2024) have also discovered the verbose problem of LLM generation, and included some length control methods to reduce the length of generated responses while minimizing the impact of LLM performances.

The previous work focused on pairwise dataset, which was more tough to gather. In comparison, binary feedback like "thumb up" and "thumb down" will be easier to gather. KTO borrowed the idea of human aversion to desired over undesired data and it can handle binary feedback successfully Ethayarajh et al. (2024). DRO focused on binary data by estimating the policy and value functions and optimize each sequentially while maintaining the other fixed Richemond et al. (2024). However, there have not been a work that can unify both pairwise and binary feedback. Nash learning model the LLM improvement as a minmax problem and propose a iterative method to gradually approach the optimal solution Munos et al. (2024). It can solve the intranstivity problem of human preference. SPPO utilized one model as two sides of the competition Wu et al. (2024). Though Nash learning provides some hints, it will increase the time of alignment as it will increase the number of iteration before convergence.

LiPO Liu et al. (2024), RRHF Yuan et al. (2023) and PRO Song et al. (2024) utilized the ranking of a list of responses, and the relative score between these methods were utilized. RPO proposd to utilize KL divergence to mimize the difference between predicted reward and labelled reward by human or AI, which is closer to our idea in this work Nvidia et al. (2024).

## 5 Conclusion

Despite the trillions of tokens used to pretrain LLMs with billions of parameters, undesired responses persist. RLHF, DPO and KTO can improve the alignment quality. However, RLHF, DPO and KTO each have their own strengths and drawbacks, but they cannot be unified into a single approach. In this work, we propose UNA to integrate the benefits of RLHF, DPO, and KTO into a unified framework. Based on the RLHF objective, the optimal policy is induced by $r(x, y) = \beta \log\left(\frac{\pi_\theta(y|x)}{\pi_{\text{ref}}(y|x)}\right) + f(x) + c$. When $f(x) = c = 0$, the reward can be simplified to $r(x, y) = \beta \log\left(\frac{\pi_\theta(y|x)}{\pi_{\text{ref}}(y|x)}\right)$. With this derived implicit reward function, it can be utilized to build UNA, which unifies RLHF, DPO and KTO as a task of minimization between implicit and explicit reward functions. As a result, UNA simplifies, stabilizes and reduces memory cost of RLHF. Downstream tasks demonstrate that UNA significantly outperforms RLHF. Then, UNA can deal with pairwise, binary and score-based feedback. For pairwise feedback, UNA is mathematically equivalent to DPO. For binary feedback, UNA can improve over KTO. For score-based feedback, UNA outperforms non-score-based methods including DPO and KTO, and it can be regarded as a distillation of RM and LLM or an offline RLAIF. In conclusion, UNA has introduced a unified, stable, and efficient approach to LLM alignment that delivers high-quality results.

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

## A  DPO: RELATIONSHIP BETWEEN OPTIMAL POLICY AND REWARD FUNCTION

The objective of RLHF / DPO is shown in Equation 3. From the objective, the relationship between optimal reward and optimal policy can be derived in Equation 4 where $Z(x) = \sum_y \pi_{\text{ref}}(y|x)e^{\left(\frac{1}{\beta}r_\theta(x,y)\right)}$. The illustration for deriving DPO is shown in Equation 14.

$$
\begin{aligned}
\pi_\theta^*(y|x) &= \max_{\pi_\theta} \mathbb{E}_{x \sim D}\left[\mathbb{E}_{y \sim \pi_\theta(y|x)} r_\theta(x,y) - \beta D_{\text{KL}}\left(\pi_\theta(y|x)\|\pi_{\text{ref}}(y|x)\right)\right] \\
&= \max_{\pi_\theta} \mathbb{E}_{x \sim D}\left\{\mathbb{E}_{y \sim \pi_\theta(y|x)}\left[r(x,y) - \beta \log \frac{\pi_\theta(y|x)}{\pi_{\text{ref}}(y|x)}\right]\right\} \\
&= \min_{\pi_\theta} \mathbb{E}_{x \sim D}\left\{\mathbb{E}_{y \sim \pi_\theta(y|x)}\left[\log \frac{\pi_\theta(y|x)}{\pi_{\text{ref}}(y|x)} - \frac{1}{\beta}r(x,y)\right]\right\} \\
&= \min_{\pi_\theta} \mathbb{E}_{x \sim D}\left\{\mathbb{E}_{y \sim \pi_\theta(y|x)}\left[\log\left(\frac{\pi_\theta(y|x)}{\frac{1}{Z(x)}\pi_{\text{ref}}(y|x)e^{\frac{1}{\beta}r(x,y)}}\right) - \log\left(Z(x)\right)\right]\right\} \\
&= \min_{\pi_\theta} \mathbb{E}_{x \sim D}\left\{\mathbb{E}_{y \sim \pi_\theta(y|x)}\left[\log\left(\frac{\pi_\theta(y|x)}{\frac{1}{Z(x)}\pi_{\text{ref}}(y|x)e^{\frac{1}{\beta}r(x,y)}}\right)\right] - \log\left(Z(x)\right)\right\} \\
&= \min_{\pi_\theta} \mathbb{E}_{x \sim D}\left\{D_{KL}\left(\pi_\theta(y|x)\|\frac{1}{Z(x)}\pi_{\text{ref}}(y|x)e^{\frac{1}{\beta}r(x,y)}\right) - \log\left(Z(x)\right)\right\}
\end{aligned}
\tag{14}
$$

The objective function is minimized when $D_{KL}\left(\pi_\theta(y|x)\|\frac{1}{Z(x)}\pi_{\text{ref}}(y|x)e^{\frac{1}{\beta}r(x,y)}\right) = 0$, and this is equivalent to $\pi_\theta(y|x) = \frac{1}{Z(x)}\pi_{\text{ref}}(y|x)e^{\frac{1}{\beta}r(x,y)}$. By rewriting, the reward model can be expressed in term of the current policy as shown in Equation 4.

However, the term $Z(x)$ cannot be computed as it needed to be computed by summing all candidate responses $y$. DPO avoids this problem by subtracting the rewards of desired and undesired responses $r(x, y_w) - r(x, y_l) = \beta\left[\log\left(\frac{\pi_\theta(y_w|x)}{\pi_{\text{ref}}(y_w|x)}\right) - \log\left(\frac{\pi_\theta(y_l|x)}{\pi_{\text{ref}}(y_l|x)}\right)\right]$. In addition, the authors argue "We say that two reward functions $r(x, y)$ and $r'(x, y)$ are equivalent iff $r(x, y) - r'(x, y) = f(x)$ for some function $f$". However, rigorous proof cannot be provided and it is only provided that $r(x, y)$ and $r'(x, y)$ induce the same optimal policy. For Lipo, $r(x, y) = \beta \log\left(\frac{\pi_\theta(y|x)}{\pi_{\text{ref}}(y|x)}\right)$ is directly utilized as rewards for listwise responses and KTO estimates $Z(x)$ by averaging over multiple samples.

## B    Mathematical Proof of UNA

In the section, the mathematical proof of UNA will be provided. For the proof of how to derive the mapping of optimal policy and reward model in DPO can be found in appendix A. Inspired by the proof of DPO, we will **rigorously prove** that $r(x,y) = \beta \log \left( \frac{\pi_\theta(y|x)}{\pi_{\text{ref}}(y|x)} \right) + f(x) + c$ will maximize the objective in Equation 3 and $r(x,y) = \beta \log \left( \frac{\pi_\theta(y|x)}{\pi_{\text{ref}}(y|x)} \right)$ is the simplest reward with $f(x) = c = 0$.

**Proposition 1.** Let $a_1, \ldots, a_n$ and $b_1, \ldots, b_n$ be non-negative numbers. Denote the sum of all $a_i$ by $a$ and the sum of all $b_i$ by $b$. The log sum inequality states Equation 15 with equality if and only if $\frac{a_i}{b_i}$ are equal for all $i$, in other words $a_i = \lambda \times b_i$ for all $i$. The proof could be found in C

$$\sum_{i=1}^{n} a_i \log \frac{a_i}{b_i} \geq a \log \frac{a}{b} \tag{15}$$

Starting from the same objective in Equation 3, it can be simplified as shown in Equation 16.

$$
\begin{aligned}
\pi_\theta^*(y|x) &= \max_{\pi_\theta} \mathbb{E}_{x \sim D} \left[ \mathbb{E}_{y \sim \pi_\theta(y|x)} r_\theta(x,y) - \beta D_{\text{KL}} \left( \pi_\theta(y|x) \| \pi_{\text{ref}}(y|x) \right) \right] \\
&= \max_{\pi_\theta} \mathbb{E}_{x \sim D} \left\{ \mathbb{E}_{y \sim \pi_\theta(y|x)} \left[ r(x,y) - \beta \log \frac{\pi_\theta(y|x)}{\pi_{\text{ref}}(y|x)} \right] \right\} \\
&= \beta \max_{\pi_\theta} \mathbb{E}_{x \sim D} \left\{ \mathbb{E}_{y \sim \pi_\theta(y|x)} \left[ \frac{1}{\beta} r(x,y) - \log \frac{\pi_\theta(y|x)}{\pi_{\text{ref}}(y|x)} \right] \right\} \\
&= \beta \max_{\pi_\theta} \mathbb{E}_{x \sim D} \left\{ \mathbb{E}_{y \sim \pi_\theta(y|x)} \left[ - \log \left( \frac{\pi_\theta(y|x)}{\pi_{\text{ref}}(y|x) e^{\frac{1}{\beta} r(x,y)}} \right) \right] \right\} \\
&= \beta \max_{\pi_\theta} \mathbb{E}_{x \sim D} \left\{ \mathbb{E}_{y \sim \pi_\theta(y|x)} \left[ - \log \left( \frac{\pi_\theta(y|x)}{\pi_{\text{ref}}(y|x) e^{\frac{1}{\beta} (r(x,y) - f(x))}} \right) + \frac{1}{\beta} f(x) \right] \right\} \\
&= \beta \max_{\pi_\theta} \mathbb{E}_{x \sim D} \left\{ \mathbb{E}_{y \sim \pi_\theta(y|x)} \left[ - \log \left( \frac{\pi_\theta(y|x)}{\pi_{\text{ref}}(y|x) e^{\frac{1}{\beta} (r(x,y) - f(x))}} \right) \right] + \frac{1}{\beta} f(x) \right\}
\end{aligned} \tag{16}
$$

Based on the log-sum inequality in Equation 15, the term can be further simplified as shown in Equation 17 because both $\pi_\theta(y|x)$ and $\pi_{\text{ref}}(y|x) e^{\frac{1}{\beta}(r(x,y)-f(x))}$ are non-negative.

$$
\begin{aligned}
&\beta \mathbb{E}_{x \sim D} \left\{ \mathbb{E}_{y \sim \pi_\theta(y|x)} \left[ - \log \left( \frac{\pi_\theta(y|x)}{\pi_{\text{ref}}(y|x) e^{\frac{1}{\beta}(r(x,y)-f(x))}} \right) \right] + \frac{1}{\beta} f(x) \right\} \\
&= \beta \mathbb{E}_{x \sim D} \left\{ - \sum_y \left[ \pi_\theta(y|x) \log \left( \frac{\pi_\theta(y|x)}{\pi_{\text{ref}}(y|x) e^{\frac{1}{\beta}(r(x,y)-f(x))}} \right) \right] + \frac{1}{\beta} f(x) \right\} \\
&\leq \beta \mathbb{E}_{x \sim D} \left\{ \left[ - \left( \sum_y \pi_\theta(y|x) \right) \log \left( \frac{\sum_y \pi_\theta(y|x)}{\sum_y \pi_{\text{ref}}(y|x) e^{\frac{1}{\beta}(r(x,y)-f(x))}} \right) \right] + \frac{1}{\beta} f(x) \right\} \\
&= \beta \mathbb{E}_{x \sim D} \left\{ \left[ -1 \log \left( \frac{1}{\sum_y \pi_{\text{ref}}(y|x) e^{\frac{1}{\beta}(r(x,y)-f(x))}} \right) \right] + \frac{1}{\beta} f(x) \right\} \\
&= \beta \mathbb{E}_{x \sim D} \left\{ \log \left( \mathbb{E}_{y \sim \pi_{\text{ref}}(y|x)} e^{\frac{1}{\beta}(r(x,y)-f(x))} \right) + \frac{1}{\beta} f(x) \right\}
\end{aligned} \tag{17}
$$

As a result, the maximum value of the objective function $\max_{\pi_\theta} \mathbb{E}_{x \sim D} \left[ \mathbb{E}_{y \sim \pi_\theta(y|x)} r_\theta(x,y) - \beta D_{\text{KL}} \left( \pi_\theta(y|x) \| \pi_{\text{ref}}(y|x) \right) \right]$ in eq 16 is

$\beta \mathbb{E}_{x \sim D} \left\{ \log \left( \mathbb{E}_{y \sim \pi_{\text{ref}}(y|x)} e^{\frac{1}{\beta}(r(x,y)-f(x))} \right) + \frac{1}{\beta} f(x) \right\}$ in Equation 17, and this inequality reaches the equality condition when Equation 18 is satisfied where $\lambda$ is a constant.

$$\frac{\pi_\theta(y|x)}{\pi_{\text{ref}}(y|x) e^{\frac{1}{\beta}(r(x,y)-f(x))}} = \frac{1}{\lambda} \tag{18}$$

By rewriting this term, we can obtain the reward in term of the policy as shown in Equation 19. In special case, $f(x) = c = 0$, it is simplified to $r(x,y) = \beta \log \left( \frac{\pi_\theta(y|x)}{\pi_{\text{ref}}(y|x)} \right)$. The condition $f(x) = c = 0$ refers that implicit and explicit reward models are exactly the same.

$$\begin{aligned} r(x,y) &= \beta \log \left( \frac{\lambda \pi_\theta(y|x)}{\pi_{\text{ref}}(y|x)} \right) + f(x) \\ &= \beta \log \left( \frac{\pi_\theta(y|x)}{\pi_{\text{ref}}(y|x)} \right) + f(x) + \beta \log(\lambda) \\ &= \beta \log \left( \frac{\pi_\theta(y|x)}{\pi_{\text{ref}}(y|x)} \right) + f(x) + c \text{ when } c = \beta \log(\lambda) \end{aligned} \tag{19}$$

When plugging Equation 18 in Equation 17, the upper bound can be simplified into a constant $\beta \log(\lambda) + \mathbb{E}_{x \sim D}(f(x))$ as shown in Equation 20.

$$\begin{aligned} & \beta \mathbb{E}_{x \sim D} \left\{ \log \left( \mathbb{E}_{y \sim \pi_{\text{ref}}(y|x)} e^{\frac{1}{\beta}(r(x,y)-f(x))} \right) + \frac{1}{\beta} f(x) \right\} \\ &= \beta \mathbb{E}_{x \sim D} \left\{ \log \left( \mathbb{E}_{y \sim \pi_{\text{ref}}(y|x)} \frac{\lambda \pi_\theta(y|x)}{\pi_{\text{ref}}(y|x)} \right) + \frac{1}{\beta} f(x) \right\} \\ &= \beta \mathbb{E}_{x \sim D} \left\{ \log \left( \mathbb{E}_{y \sim \pi_\theta(y|x)} \lambda \right) + \frac{1}{\beta} f(x) \right\} \\ &= \beta \mathbb{E}_{x \sim D} \left\{ \log(\lambda) + \frac{1}{\beta} f(x) \right\} \\ &= \beta \log(\lambda) + \mathbb{E}_{x \sim D}(f(x)) \end{aligned} \tag{20}$$

When desired to generalize this into "infinite dimension", another constraint needs to be added, i.e., $\sum_y \pi_{\text{ref}}(y|x) e^{\frac{1}{\beta}(r(x,y)-f(x))}$ should be finite. Then, $f(x)$ is further restricted to $f(x) > \max[r(x,y)]$ with normalization on $r(x,y)$ in advance. Eventually, $\sum_y \pi_{\text{ref}}(y|x) e^{\frac{1}{\beta}(r(x,y)-f(x))} < \sum_y \pi_{\text{ref}}(y|x) = 1$, which will be finite.

Here is a brief summary of this section, based on this objective $\pi_\theta^*(y|x) = \max_{\pi_\theta} \mathbb{E}_{x \sim D} \left[ \mathbb{E}_{y \sim \pi_\theta(y|x)} r_\theta(x,y) - \beta D_{\text{KL}} \left( \pi_\theta(y|x) \| \pi_{\text{ref}}(y|x) \right) \right]$ in Equation 3, we can obtain its upper bound $\beta \mathbb{E}_{x \sim D} \left\{ \log \left( \mathbb{E}_{y \sim \pi_{\text{ref}}(y|x)} e^{\frac{1}{\beta}(r(x,y)-f(x))} \right) + \frac{1}{\beta} f(x) \right\}$ as shown in Equation 17. The upper bound, i.e., the equality condition is reached when $r(x,y) = \beta \log \left( \frac{\pi_\theta(y|x)}{\pi_{\text{ref}}(y|x)} \right) + f(x) + c$ as shown in Equation 19. It can be further simplified to $r(x,y) = \beta \log \left( \frac{\pi_\theta(y|x)}{\pi_{\text{ref}}(y|x)} \right)$ if $f(x) = c = 0$. In particular, $f(x) = c = 0$ refers the implicit reward equals to explicit rewards. Lastly, when the equality condition is reached, the upper bound would be $\beta \log(\lambda) + \mathbb{E}_{x \sim D}(f(x))$ as shown in Equation 20.

## C DERIVATION OF LOG-SUM INEQUALITY

**Jesen inequality.** For a real convex function $\varphi$, numbers $x_1, x_2, \ldots, x_n$ in its domain, and positive weights $a_i$, Jensen's inequality can be stated as in Equation 21:

$$\frac{\sum_{i=1}^{n} a_i \varphi(x_i)}{\sum_{i=1}^{n} a_i} \geq \varphi \left( \frac{\sum_{i=1}^{n} a_i x_i}{\sum_{i=1}^{n} a_i} \right) \tag{21}$$

**Proof of log-sum inequality.** Firstly, define $f(x) = x \log(x)$. Then, $f'(x) = 1 + \log(x)$ and $f''(x) = \frac{1}{x}$. For the domain $x > 0$, $f''(x) > 0$. As a result, $f(x) = x \log(x)$ is a concvex function and satisfy Jesen's inequality. Then, the log-sum inequality could be derived in Equation 22.

$$\begin{aligned}
\sum_{i=1}^{n} a_i \log \left( \frac{a_i}{b_i} \right) &= \sum_{i=1}^{n} b_i f \left( \frac{a_i}{b_i} \right) \\
&= b \sum_{i=1}^{n} \frac{b_i}{b} f \left( \frac{a_i}{b_i} \right) \\
&= b \frac{\sum_{i=1}^{n} b_i f \left( \frac{a_i}{b_i} \right)}{\sum_{i=1}^{n} b_i} \\
&\geq b f \left[ \frac{\sum_{i=1}^{n} b_i \frac{a_i}{b_i}}{\sum_{i=1}^{n} b_i} \right] \\
&= b f \left( \frac{a}{b} \right)
\end{aligned} \tag{22}$$

## D    DEFAULT NOTATION

$x$**:** prompt to LLM

$y_w$ : desired response

$y_l$ : undesired response

$P(y_w > y_l|x)$ : the probability of desired response over undesired response

$r_\phi(x, y)$ : the explicit reward

$r_\theta(x, y)$ : the implicit reward

$s_\phi(x, y)$ : the explicit score: normalized explicit reward

$s_\theta(x, y)$ : the implicit score: normalized implicit reward

$D_{KL}$ : KL divergence

$\pi_\theta$ : LLM policy to be aligned

$\pi_{ref}$ : reference policy for LLM alignment

$g(\cdot)$ : any function that measures the difference between implicit and explicit reward functions

