# OpenReview forum: "UNA: Unifying Alignments of RLHF/PPO, DPO and KTO by a Generalized Implicit Reward Function"
_ICLR.cc/2025/Conference — ICLR 2025 Conference Withdrawn Submission_

### Official Review · Reviewer_qkms · 2024-10-16

**Soundness:** 2
**Presentation:** 1
**Contribution:** 2
**Rating:** 3
**Confidence:** 4

**Summary:**

This paper introduces a method called Unified Alignment (UNA), which aims to unify existing alignment techniques like RLHF, DPO, and KTO. The authors argue that RLHF, though effective, suffers from high complexity and instability, while DPO simplifies the process but is limited in certain aspects, such as not fully leveraging the reward model and being constrained to pairwise preference data. The proposed UNA framework claims to generalize and improve upon these methods by creating a novel mapping between reward models and optimal policies, allowing it to handle different feedback types and improve performance across downstream tasks. The paper suggests that UNA can simplify, stabilize, and speed up fine-tuning processes while reducing memory requirements, and it reports outperforming RLHF, DPO, and KTO in experiments.

**Strengths:**

It is of interest that the authors try to narrow the gap between the RM-based and RM-free alignment methods.

**Weaknesses:**

The paper is not solid enough for me and the contribution is limited. The reasons are concluded as follow:

First, It is not clear to me why the proposed method differs from DPO, thus the first contribution claimed is not novel at all. In my understanding, the DPO has already proved the equivalent between the optimal policy and the reward model's objective (under the BT model setting). Besides, it is also very confusing why the authors introduce f(x) and c.

Second, for the claimed contribution 2, the author used a objective introduced by DPO model to conduct a supervised learning, which is also confusing. What is the difference between using the proposed method to train an additional model and directly using a well-trained DPO model? And what if the DPO model is not well-trained, then the label would be very noisy? It is a very confusing step and I am open to more discussion.

Third, the proposed method is compared to some baselines and is claimed to be better, however I believe the comparisons are not fair and the experimental results are suspicious.  In Table 1, the authors use 4 UNA variants and only use KTO and Mistral as the baselines, and the best scores in each benchmark are distributed to all 4 variants, so which one is the best? The authors also claim UNA perform better than RLHF while they didn't provide the experimental detail about RLHF. The proposed method is not even a RL algorithm since it doesn't shown any potential in exploration but more like a distillation process, it is meaningless to announce that it perform better than PPO regarding speeds or hardware consuming.

The presentation of the paper needs to be largely improved. There are many compile Errors such as line 215 in the 4th page. Given all these reasons, I will not recommend the paper to be accepted.

**Questions:**

Please refer to weakness part.

---

> ### Author Response · Authors · 2024-11-16
>
> 1. We have proved that in the appendix.
>
> 2. DPO can only deal with pairwise data (a > b), KTO can deal with binary data (0 and 1) and UNA can deal with different types of data (pair, binary and feedback).
>
> 3. UNA-binary is using binary feedback (0 or 1) and UNA-score is using score feedback (0 to 1). In RLHF, we follow huggingface RLHF and reports its result.

---

> ### Comment · Reviewer_qkms · 2024-11-26
>
> Thank the authors for the feedback, but it is still a confusion to me why RLHF is not considered in all benchmarks and experiments reported as baselines. If the author can show additional results or give reasonable explanations, I would consider raise my score.

---

> > ### Author Response · Authors · 2024-12-03
> >
> > RLHF is online alignment, while DPO and KTO is an offline alignment. UNA can be both online and offline. We compare offline UNA with DPO and KTO in table 1 and 2. We compare online UNA with RLHF in table 3 and 4.

---

### Official Review · Reviewer_9KnH · 2024-11-02

**Soundness:** 1
**Presentation:** 1
**Contribution:** 1
**Rating:** 3
**Confidence:** 4

**Summary:**

This paper presents an algorithm which they call UNified Alignment (UNA), by utilizing a simplified DPO implicit reward. UNA's loss minimizes the difference between the implicit reward and the explicit reward (provided by an existing reward model), hence can be generalized to different modes of feedback: pairwise, binary or scalar. Empirical results are presented to compare UNA with several baselines.

**Strengths:**

1. There are math derivations for the proposed method.
2. The reward distillation loss is  general for different modes of feedback.

**Weaknesses:**

W1. The paper's presentation does not meet the standard of ICLR.
* Some references are missing (215,353); there are grammar and spelling errors (026,106,120,141).
* Figures 1 and 2 are almost identical except for the different parts highlighted, which makes me feel they are redundant. You could refine the two figures and make them respectively focused on the topic you want to highlight: algorithm comparison & applications to different modes of feedback.
* The overall tune is not formal, and the writing is not easy to follow (too many pieces). If your goal is to develop a unified framework for LLM alignment, in your methodology you would start from defining the overall process of alignment, then the objectives of it, followed by your approach. Maybe at the end you can make connections with existing methods like RLHF with PPO, DPO, KTO and say they are some special cases. This suggestion is only one possible way for improvement. There are certainly more that you could consider to improve the current version.

W2. The technical novelty is limited. The implicit reward (eq6) of DPO is standard and well-known. One of the contributions is "simplifying" the implicit reward by setting f(x)=c=0. Could you provide insights behind it? Besides, the other contribution is a supervised loss of minimizing the difference between implicit and explicit rewards (eq7) is also not novel. For example, [1] proposed such a distillation-style loss for robust preference optimization.


[1] Fisch A, Eisenstein J, Zayats V, Agarwal A, Beirami A, Nagpal C, Shaw P, Berant J. Robust preference optimization through reward model distillation. arXiv preprint arXiv:2405.19316. 2024 May 29.

**Questions:**

It seems to me UNA also needs a reward model to learn the policy, which is not too different from RLHF. But in your abstract, "RLHF requires training the reward model and policy separately, which is complex, time-consuming, memory intensive and unstable during training processes." Can you clarify the advantage of UNA compared to standard RLHF? For example, it could be that UNA saves some computation, or converges faster, etc.

---

### Official Review · Reviewer_cBvy · 2024-11-03

**Soundness:** 3
**Presentation:** 1
**Contribution:** 3
**Rating:** 5
**Confidence:** 4

**Summary:**

This paper provides a unifying framework for value alignment that has:
(i) explicit and implicit reward functions
(ii) can handle different types of preferences (pairwise, binary, score etc.)
and seems to provide better performance than state of the art approaches, RLHF, DPO, KTO.

**Strengths:**

1. The unified framework does seem to have significant advantages.
2. The paper's ideas are strong.
3. Experimental results are detailed and strong.

**Weaknesses:**

1. The biggest concern I have is that presentation is extremely poor, with many typos in notation, proofs, text and references.
2. There are notational inconsistencies in proofs as well.
3. There is no intuitive explanation for why such a unified approach would provide better performance than any of the individual approaches.

I find that the paper is written in a rushed manner and even though the ideas are interesting, it does not seem ready yet with typos all over the paper. Happy to revise my scores based on the responses of the authors.

**Questions:**

1. What is the intuitive explanation for why a unified approach provides better results? DPO provides better results than RLHF as it does direct optimization of preferences, rather than first creating a reward function and then optimizing. In a similar vein, is there an intuitive explanation.
2. There are errors in the main proof in Appendix B. While I seem to follow the broad outline, I am not entirely sure and would request the authors to provide more explanation and a corrected version. "y" is present on the left hand side, there is a "y" used in expectation. I am not sure how the expectation over "y" makes sense?
3. Are the experimental improvements significant? The improvements are there, but are they significant? How can it be quantified?

---

> ### Author Response · Authors · 2024-11-16
>
> Weakness:
> There is no intuitive explanation for why such a unified approach would provide better performance than any of the individual approaches.
> We use more accurate information than DPO and KTO. DPO uses pairwise information (a > b), and KTO uses binary feeback (0, 1). However, UNA uses score-based information, which is more accurate than DPO and KTO.
>
> Question:
> There is no intuitive explanation for why such a unified approach would provide better performance than any of the individual approaches.
> UNA use more accurate feedbacks than DPO and KTO as stated before.
>
> DPO provides better results than RLHF as it does direct optimization of preferences, rather than first creating a reward function and then optimizing. In a similar vein, is there an intuitive explanation.
> In my opinion, DPO is not guaranteed to perform better than RLHF as DPO only utilizes pairwise information and RLHF uses score-based information. As a result, the information used in RLHF is more accurate than DPO. There have been some works that proves RLHF outperforms DPO in some tasks. The benefit of RLHF lies in the memory cost and unstability.
>
> There are errors in the main proof in Appendix B. While I seem to follow the broad outline, I am not entirely sure and would request the authors to provide more explanation and a corrected version. "y" is present on the left hand side, there is a "y" used in expectation. I am not sure how the expectation over "y" makes sense?
>
> The expectation means that given a prompt, there are multiple responses, and the expectation means that given a prompt, and multiple responses, the expectation of different responses.
>
>  Are the experimental improvements significant? The improvements are there, but are they significant? How can it be quantified?
> From table 1, the improvement of DPO over initial model is 2%, and the improvement of UNA to DPO is nearly 2%. So the improvement is large from my viewpoint.

---

### Official Review · Reviewer_kN8x · 2024-11-04

**Soundness:** 2
**Presentation:** 2
**Contribution:** 3
**Rating:** 3
**Confidence:** 4

**Summary:**

This paper introduces the UNA framework, which integrates three prominent alignment techniques for LLMs: RLHF/PPO, DPO, and KTO. The UNA framework seeks to unify these methods by using a generalized implicit reward function to align language model policies through supervised learning. UNA also accommodates diverse feedback types (pairwise, binary, and score-based) and aims to outperform each technique on downstream tasks, simplifying the RLHF fine-tuning process.

**Strengths:**

1. By combining RLHF, DPO, and KTO, UNA supports a range of feedback types (pairwise, binary, and score-based), which enhances its versatility.
2. UNA replaces RLHF’s unstable, memory-heavy RL process with supervised learning, making the alignment process more straightforward and efficient.
3. The paper provides a mathematical proof linking the RLHF objective to a generalized implicit reward function.

**Weaknesses:**

1. Evaluation of the experiments is relatively small-scale. The authors only evaluate UNA over DPO & KTO when fine-tuning with LoRA and evaluate online RLHF using a 1.5B model. It may be difficult to prove whether UNA can take advantage when using large-scale models. I suggest the authors to try larger models like ~8B models since they claim that they use 8 x 80G A100 GPUs.
2. The improvement of UNA over three baselines are not significant, which is considerate since they are somewhat equivalent. However, I think the authors should show the advantages of UNA, e.g., the time and computational cost, compared to baselines.
3. The paper seems written in a rush with several clarity issues. For example, some references (Line 215 and 352) are missing.

**Questions:**

1. The paper states that $f(x)=0$ when implicit and explicit reward models are exactly the same. However, when we optimize a model using UNA with the simplified implicit reward $f(x)=c=0$, will it guarantee that it will lead to the convergence?
2. DPO also use log pi / pi_ref as an indicator of the reward. However, this implicit reward cannot well reflect the training status. Can the UNA framework better stabilize the training process in the DPO manner?
3. In Table 3, maybe you can try other reference models rather than GPT in Alpaca-eval? The win rates against GPT-4 is to low and difficult to compare among different models.
4. Can you include the results of the original DPO?

---

> ### Author Response · Authors · 2024-11-16
>
> Weakness:
> 1. 8*80G A100 cannot support 8B policy model and reward model as LoRA is supported by Huggingface.
> 2. It is significant for two tasks ifeval and truthful-qa, which is most important for sft and alignment separately. Other tasks are not improved significantly because of the corresponding data are not utilized.
> 3. It is because I transform from the main body of the paper to the appendix.
>
> Question:
> 1. f(x)=0 is a special simplified version of the generalized implicit reward function. UNA goal is minimize the difference between implicit reward and explicit reward, where explicit reward is pairwise, binary and score feedback.
> 2. DPO can only work with pairwise data while UNA can work with any score-based feedback. DPO, KTO and UNA are generally stable as they are supervised learning.
> 3. Got it. But the open llm leaderboard should be more reliable in our viewpoint.
> 4. I have included the original results of DPO and KTO.

---

> > ### Comment · Reviewer_kN8x · 2024-11-26
> >
> > I thank the authors for their rebuttal. I would like to raise some further concerns.
> >
> > > 8*80G A100 cannot support 8B policy model and reward model as LoRA is supported by Huggingface.
> >
> > I think it will be fine if you configure deepspeed properly. Alternatively, maybe you can try use an 8B policy with a smaller reward model. But in general, only presenting 1.5B results seems not enough if you want to illustrate that UNA is comparable to DPO or KTO, as 1.5B models may fail to produce reasonable outputs in some cases.
> >
> > > f(x)=0 is a special simplified version of the generalized implicit reward function.
> >
> > Could you further elaborate on the assumptions and intuition behind this simplification?
> >
> > > I have included the original results of DPO and KTO.
> >
> > I only see the results of DPO (UNA-pairwise). Is the implementation exactly identical to DPO?

---

> > > ### Author Response · Authors · 2024-12-03
> > >
> > > Thank you for the reviewer's feedback.
> > > 1.
> > > Question: I think it will be fine if you configure deepspeed properly. Alternatively, maybe you can try use an 8B policy with a smaller reward model. But in general, only presenting 1.5B results seems not enough if you want to illustrate that UNA is comparable to DPO or KTO, as 1.5B models may fail to produce reasonable outputs in some cases.
> > > Reply: we will try to work on 8B for RLHF in the future.
> > >
> > > 2.
> > > Question: Could you further elaborate on the assumptions and intuition behind this simplification?
> > > Reply: We just discover that it is a special case of the general reward function. When this simplification is true, we can use score or more detailed evaluation for alignment.
> > >
> > > 3.
> > > Question: I only see the results of DPO (UNA-pairwise). Is the implementation exactly identical to DPO?
> > > Reply: Yes, it is.

---

### Note · Authors · 2025-01-21

I have read and agree with the venue's withdrawal policy on behalf of myself and my co-authors.